*Report*

# Exploiting native forces to capture chromosome conformation in mammalian cell nuclei

Lilija Brant[1], Theodore Georgomanolis[1,†], Milos Nikolic[1,†], Chris A Brackley[2], Petros Kolovos[3], Wilfred van Ijcken[4] (iD), Frank G Grosveld[3], Davide Marenduzzo[2] & Argyris Papantonis[1,*] (iD)

## Abstract

Mammalian interphase chromosomes fold into a multitude of loops to fit the confines of cell nuclei, and looping is tightly linked to regulated function. Chromosome conformation capture (3C) technology has significantly advanced our understanding of this structure-to-function relationship. However, all 3C-based methods rely on chemical cross-linking to stabilize spatial interactions. This step remains a "black box" as regards the biases it may introduce, and some discrepancies between microscopy and 3C studies have now been reported. To address these concerns, we developed "i3C", a novel approach for capturing spatial interactions without a need for cross-linking. We apply i3C to intact nuclei of living cells and exploit native forces that stabilize chromatin folding. Using different cell types and loci, computational modeling, and a methylation-based orthogonal validation method, "TALE-iD", we show that native interactions resemble cross-linked ones, but display improved signal-to-noise ratios and are more focal on regulatory elements and CTCF sites, while strictly abiding to topologically associating domain restrictions.

**Keywords** chromatin looping; chromosome conformation capture; cross-linking; nuclear compartments; nuclear organization

**Subject Categories** Chromatin, Epigenetics, Genomics & Functional Genomics; Genome-Scale & Integrative Biology; Methods & Resources

**Mol Syst Biol. (2016) 12: 891**

See also: **MJ Rowley & VG Corces** (December 2016)

## Introduction

The higher-order folding of mammalian chromosomes has long been linked to the regulation of their function. However, over the last decade, studies exploited 3C technology to significantly advance our understanding of this structure-to-function relationship (Dekker *et al*, 2013; Pombo & Dillon, 2015; Denker & de Laat, 2016) and allowed us to address diverse biological questions (e.g., Tolhuis *et al*, 2002; Papantonis *et al*, 2012; Zhang *et al*, 2012; Naumova *et al*, 2013; Rao *et al*, 2014). We now know that interphase chromosomes are partitioned into topologically associating domains (TADs; Dixon *et al*, 2012) ranging from 0.1 to few Mbp. TADs are rich in intradomain (versus interdomain) multi-loop interactions connecting genes and *cis*-regulatory elements, and their boundaries remain largely invariant between different cell types (Dixon *et al*, 2012; Rao *et al*, 2014) or upon cytokine signaling (Jin *et al*, 2013; Le Dily *et al*, 2014).

3C methods rely on chemical cross-linking for stabilizing and capturing spatial interactions. Although formaldehyde is widely used in molecular biology, and its chemistry is well understood, its *in vivo* effects remain obscure (Gavrilov *et al*, 2015). For instance, not all nuclear proteins/loci are equally efficiently cross-linked (Teytelman *et al*, 2013); cross-linking may trigger the DNA damage response to induce polyADP-ribosylation of the proteome and thus alter its susceptibility to fixation (Beneke *et al*, 2012), while fixation is sensitive to even slight changes in temperature, pH, and duration (Schmiedeberg *et al*, 2009). Then, it is conceivable that variations in fixation efficiency within the different nuclear microenvironments (e.g., in hetero- versus euchromatin) can skew experimental readouts. Discrepancies between DNA FISH and 5C studies in the *HoxD* locus were recently reported (Williamson *et al*, 2014). FISH requires harsher fixation than the 5C procedure and, in multiple cases, microscopy and 3C results do correlate well; nonetheless, such discrepancies may, at least in part, stem from differential fixation effects in the dense chromatin mesh within TADs. Moreover, any interactions that end up being detected via 3C assays must survive harsh detergent treatment, prolonged heating and shaking, plus unphysiological salt concentrations (Stadhouders *et al*, 2013). To address these concerns, we developed "intrinsic 3C" (i3C), a novel approach to capture chromatin conformation in living cells without a need for cross-linking. i3C exploits native forces that preserve the relative spatial positioning of chromatin fragments. We generated i3C profiles in a number of cell types and loci to investigate different features of chromatin looping in the absence of chemical fixation.

1 Center for Molecular Medicine, University of Cologne, Cologne, Germany
2 School of Physics and Astronomy, University of Edinburgh, Edinburgh, UK
3 Department of Cell Biology, Erasmus Medical Center, Rotterdam, The Netherlands
4 Biomics Department, Erasmus Medical Center, Rotterdam, The Netherlands
  *Corresponding author. Tel: +49 221 478 96987; E-mail: argyris.papantonis@uni-koeln.de
  †These authors contributed equally to this work

# Results and Discussion

We apply i3C (overview in Fig 1A) to intact nuclei from living, uncross-linked, cells harvested in a buffer (PB) that closely approximates physiological salt concentrations and deters aggregation of nuclear components (Kimura *et al*, 1999; Melnik *et al*, 2011). Thus, > 95% transcriptional activity is retained (Appendix Fig S1A), suggesting that chromatin organization is also maintained. Nuclei are treated with a restriction endonuclease for ~30 min at a suboptimal temperature (not all enzymes work equally well in PB; Appendix Fig S1B), washed, and spun to remove any unattached chromatin. This removes > 40% of total chromatin (Fig 1B) and so reduces the fraction of "bystander/baseline" ligations (Dekker *et al*, 2013) to improve signal quality. Cohesive DNA ends are then ligated within intact nuclei, where native interactions are inherently preserved (Gavrilov *et al*, 2013; Rao *et al*, 2014), before the i3C template is purified (see Materials and Methods for details). The i3C workflow takes place in a single tube to minimize material losses, is faster than the conventional one (Stadhouders *et al*, 2013) and just as efficient (Appendix Fig S1C). In principle, i3C can also be applied to solid tissue (e.g., mouse liver) so long as single nuclei can be obtained.

To ensure that ligation occurs exclusively within single nuclei under native conditions, we mixed an equal number of human endothelial (HUVEC) and mouse embryonic stem cell (mESC) nuclei, performed i3C, and sequenced the resulting ligation products; < 0.7% of the reads pairs mapped one end to the human and the other end to the mouse genome. Then, as a substantial amount of DNA is lost in i3C, we asked whether any biases arise during cutting (which does not trigger the DNA damage response; Appendix Fig S1D). We treated nuclei with *Nla*III, isolated DNA at different stages along the procedure and sequenced it (steps 2–4, Fig 1A). Read profiles from "lost" (step 3) and "retained" fractions (step 4) overlap (~70% *Nla*III fragments have reads in both fractions) are equally enriched in active and inactive loci (e.g., at enhancers, CTCF–CTCF loops, lamin-associated domains; Fig 1C and Appendix Fig S1E–I), and display very similar fragment size contents (Appendix Fig S1J). Moreover, cutting chromatin in the presence or absence of cross-linking does not yield very different profiles, nor does it display a preference for "open" chromatin (despite the short incubation times used; Appendix Fig S2). Thus, the different chromatin regions are equally represented in i3C ligations.

Next, we produced an i3C template in HUVECs to query using qPCR (i3C-qPCR) interactions seen by conventional 4C for the *EDN1* housekeeping gene (Diermeier *et al*, 2014). We paired an "anchor" primer at the *EDN1* TSS with tandem primers at eight *Apo*I fragments within its locus and faithfully recapitulated all major interactions (with no interactions in the "lost" chromatin fraction; Appendix Fig S3). Some interacting fragments were separated by as many as 500 kbp, encouraging us to apply i4C-seq to our model *SAMD4A* locus (Diermeier *et al*, 2014). i4C templates were produced in HUVECs by cutting with *Apo*I, recutting and circularizing, amplifying fragments contacted by the *SAMD4A* TSS by inverse PCR, and amplimer sequencing. In parallel, the same viewpoint and primers were used to generate conventional 4C profiles. The resulting data were processed via "fourSig" (Williams *et al*, 2014) to correct for mapping biases and identify significant interactions. Comparison of i4C and conventional 4C *SAMD4A cis*-interactions

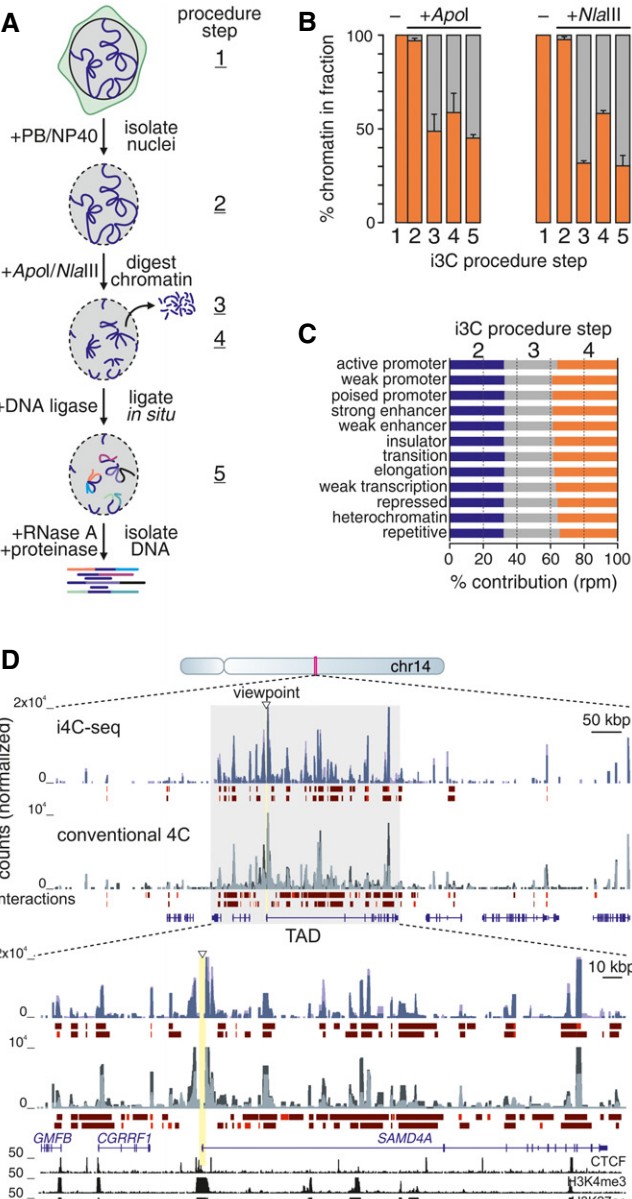

**Figure 1.   Features of i3C performed in HUVECs.**

A   Overview of the i3C protocol. Living cells are harvested in a close-to-physiological buffer (PB; step 1); intact nuclei isolated by mild NP-40 treatment (step 2); chromatin digested using *Apo*I or *Nla*III, nuclei spun to release unattached chromatin (step 3); and leave cut chromatin bound to the nuclear substructure (step 4). Then, ligation takes places *in situ*, and DNA is isolated (step 5).

B   Percentage of total cell chromatin present at the different steps of the procedure (± SD; *n* = 2).

C   Relative contribution of the different HUVEC ChromHMM features in each i3C fraction.

D   i4C-seq (blue shades) and conventional 4C (gray shades) were performed side by side in HUVECs, using *Apo*I and the *SAMD4A* TSS as a viewpoint (triangle); profiles from two replicates are overlaid. The browser view shows interactions in the ~1 Mbp around *SAMD4A*. The zoom-in shows interactions in the *SAMD4A* TAD (gray rectangle). Strong (red) and intermediate (brown) interactions called by *fourSig*, RefSeq gene models, and ENCODE ChIP-seq data are shown below.

    

revealed extensive similarities, especially within the viewpoint's TAD (Fig 1D). Few contacts were seen only in the absence of cross-linking (which was also confirmed by a differential analysis via "FourCSeq"; Klein *et al*, 2015; Appendix Fig S4). Compared to high-resolution Hi-C (Rao *et al*, 2014) and ChIA-PET contact maps (Papantonis *et al*, 2012) from HUVECs, i4C showed matching interaction profiles and aligned well within TAD boundaries, while also reproducibly detecting some longer-range contacts (Appendix Fig S5A and B). However, i4C profiles were more enriched in interactions with *cis*-regulatory regions (e.g., enhancers, CTCF sites; Appendix Fig S5C–E), and interactions unique to i4C (~65% of all *cis*-contacted fragments in i4C and conventional 4C overlap) are with genomic regions carrying the expected histone marks (e.g., H3K4me1/2, H3K36me3, H3K9ac; Appendix Fig S5F). Importantly, the signal at i4C contacts is more focal and allows deconvolution of single interactions (Fig 1D and Appendix Fig S6A and B).

Next, setting a "background" threshold at 100 rpm (as both positions and enrichments below this threshold showed the most variance in our replicates), we found 83% of i4C-seq reads over the threshold compared to < 60% of conventional 4C reads (Appendix Fig S7A). Moreover, i4C displayed significantly lower numbers of uncut and self-ligation reads for the *SAMD4A* viewpoint (a trend associated with milder fixation; van de Werken *et al*, 2012), as well as more reads mapping within its TAD (Appendix Fig S7B). This held also true for i4C from the *BMP4*, *CDKN3*, and *CNIH* genes that reside in TADs of different sizes; all displayed > 40% reads mapping within their respective TAD (Fig 2 and Appendix Fig S7C), suggesting TADs impose strong topological restrictions under native conditions.

Similar *SAMD4A* i4C profiles were also obtained in a different cell type (IMR-90) or when *Apo*I was replaced by *Nla*III (Appendix Fig S8). However, the *SAMD4A* locus is densely populated by genes and *cis*-elements. Hence, we also applied i4C to the gene-poor *EDN1* and the hetero-chromatinized *TBX5* loci. For the *EDN1* TSS viewpoint, we essentially only record i4C contacts to other active promoters and *cis*-elements, again markedly more focal and enriched for relevant chromatin marks compared to conventional data (Appendix Fig S9), while *TBX5* interacts with other H3K27me3-bound regions, including the neighboring, inactive, *TBX3* locus (Appendix Fig S10). In addition, we could reproduce previously recorded interactions at and between the *Nanog* and *Sox2* loci in mESCs (de Wit *et al*, 2013; Appendix Fig S11), confirming that i4C also captures *trans*-interactions. Finally, one tends to think that methods not using fixation require large numbers of primary material, but we typically use 5 million cells for i4C (below what is recommended for conventional 4C; Stadhouders *et al*, 2013). We also tested increasingly lower cell counts in i4C of the *SAMD4A* viewpoint; similar *cis*-profiles were obtained with < $10^6$ cells, albeit at the expense of signal-to-noise ratios (Appendix Fig S12).

Recently, *in situ* Hi-C was performed in uncross-linked lymphoblasts (typically by embedding cells in agar); despite their relative sparsity, these profiles largely matched those obtained using cross-linking (Rao *et al*, 2014). To compare this and different experimental conditions to i3C, we generated interaction profiles for the *SAMD4A* TSS. Omitting formaldehyde fixation from the *in situ* protocol results in a markedly de-enriched interactome; for instance, the *SAMD4A* TSS is looped to a cluster of enhancers in its first intron—this interaction is significantly diminished when conventional 4C is performed without cross-linking, and essentially lost once cells are treated with RNase A (Appendix Fig S13A–C). Similarly, we previously used 3C-PCR to probe interactions between DNA fragments attached to isolated transcription factories (Melnik *et al*, 2011). We incorporated the factory isolation step (using Group-III caspases) in i3C and produced "factory 4C" data for

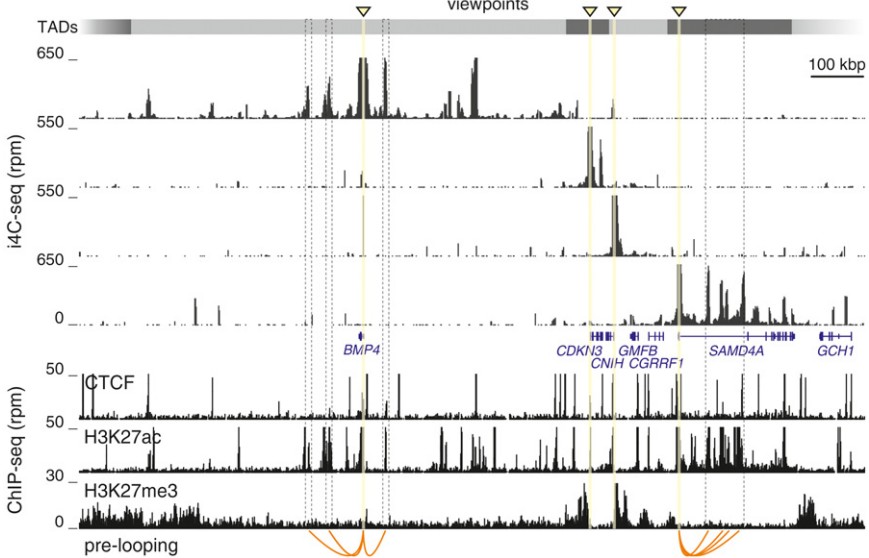

**Figure 2.  Native interactions are confined by TAD boundaries and describe prelooping.**
i4C-seq was performed in HUVECs using *Nla*III and the TSSs of *BMP4*, *CDKN3*, *CNIH*, and *SAMD4A* as viewpoints (triangles). Interactions are shown aligned to TAD boundaries (gray rectangles; from Dixon *et al*, 2012) and HUVEC ENCODE ChIP-seq data (below). Prelooping of the *SAMD4A* and *BMP4* TNF-responsive TSSs to enhancers is indicated (orange lines).

*SAMD4A*; as expected, interactions are largely preserved, but such "factory 4C" can suffer from a bias for active gene interactions, as well as from the unknown effects of caspase digestion (Appendix Fig S13D). Then, native interactions seem best detected using i3C (which avoids SDS treatment, caspase digestion, or heating) and RNA, like that produced at the *SAMD4A* enhancer cluster, may stabilize particular interactions and reduce the release of cut fragments from the nuclear substructure (Appendix Fig S13E).

Next, we used a predictive polymer modeling approach that can faithfully reproduce spatial chromatin organization based on ENCODE ChIP-seq and ChromHMM data (Brackley *et al*, 2016a) to simulate the interactome of the ~2.8-Mbp locus shown in Fig 2. We generated 500 *in silico* conformations at 1-kbp resolution, from which average simulated interaction profiles were obtained and compared to experimental 4C/i4C data (see Appendix Supplementary Methods and Appendix Fig S14A and B). In agreement with all other comparisons, i4C and conventional 4C profiles closely resemble simulated ones (e.g., *BMP4* i4C shows a correlation of 0.697 to the simulations, and 4C one of 0.745; Appendix Fig S14C).

We also devised "TALE-iD", a new orthogonal method for validating i4C interactions, as we sought to avoid FISH approaches, which require cross-linking (Williamson *et al*, 2014). We fused a custom TAL-effector DNA-binding domain (that specifically binds an enhancer in the first intron of *ZFPM2*; Mendenhall *et al*, 2013) with a bacterial adenosine methylase (Dam; Vogel *et al*, 2006). Once the targeted enhancer is found in physical proximity to other

genomic regions (due to looping), the Dam methylase will methylate adenine residues thereon (Fig 3A). This construct was introduced into K562 cells, where the target enhancer is active and i4C finds it looped to the *ZFPM2* TSS (Fig 3B). Genomic DNA from transfected K562 was then isolated and digested using *Dpn*I (that cuts only methylated sites). Cutting efficiency at 12 different sites was quantified by qPCR and showed that the targeted enhancer contacts the TSS ("p1–p4") and an upstream enhancer ("m1"). However, another enhancer further downstream is not contacted, and no interactions are detected when a "ΔDam" construct is used (Fig 3B and C).

We now understand that the promoters of stimulus-inducible genes are often prelooped to cognate enhancers (Jin *et al*, 2013). This motivated us to examine whether prelooping might arise as a result of cross-linking within tightly packed TADs. First, we applied i3C-qPCR to the *IL1A* TNF-responsive locus in HUVECs and verified prelooping under native conditions (Appendix Fig S15). We next generated i4C data for the TSSs of four genes in the same locus following a 60-min TNF pulse. Of these, the TNF-responsive *BMP4* and *SAMD4A* are prelooped to H3K27ac-decorated enhancers (Fig 2 and Appendix Fig S16). We also reasoned that the focal i4C contacts can be used to track dynamic changes in interactions upon TNF stimulation. We compared i4C and 4C profiles before and after stimulation to find more changes in the absence of cross-linking (Appendix Figs S16 and S17). For *SAMD4A*, the loops between its TSS and the downstream enhancer cluster are partially remodeled

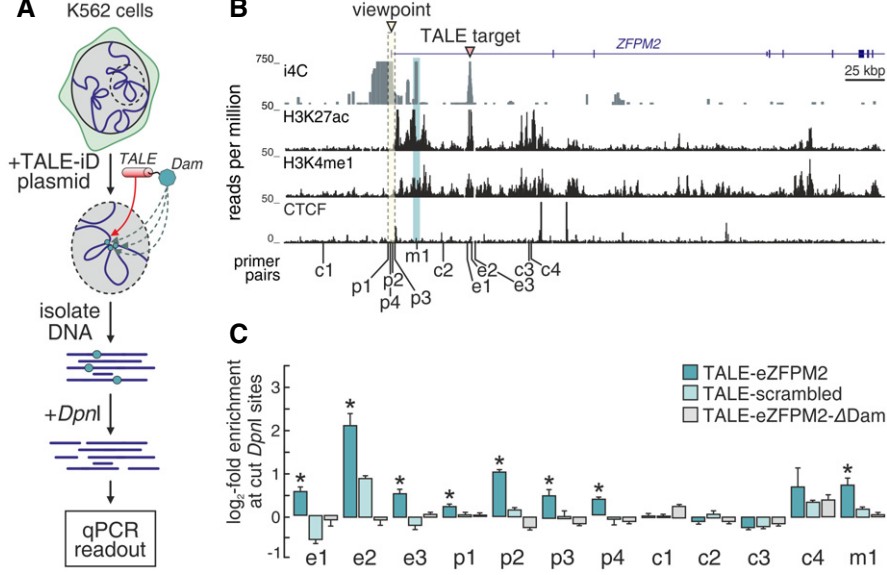

**Figure 3.  TALE-iD verifies native looping at the human *ZFPM2* locus.**

A   An overview of TALE-iD. A construct encoding a TALE DNA-binding domain that targets an active enhancer in the *ZFPM2* first intron is fused to a bacterial Dam methylase and introduced into K562 cells. Cells are harvested 48 h after transfection; genomic DNA is isolated and digested with *Dpn*I to reveal sites specifically methylated by the Dam activity. Finally, qPCR using primers flanking different *Dpn*I sites is used as readout.

B   i4C performed in K562 cells using *Apo*I and the *ZFPM2* TSS as a viewpoint (triangle). i4C interaction in the 458-kbp *ZFPM2* locus is shown, and the enhancer targeted by the TALE-iD construct is indicated (red triangle). K562 ENCODE ChIP-seq data are also shown below.

C   qPCR readout at different *Dpn*I sites. *Dpn*I sites at the *ZFPM2* promoter (p1–p4) and enhancer (e1–e3; positions in panel B) were targeted in qPCRs after restriction digest. Bar plots show $\log_2$-fold enrichment of cut sites ($1/\Delta\Delta C_t$) over background *Dpn*I cutting levels in untransfected K562 cells. Regions c1–c4 serve as controls; region m1 (an enhancer shown to interact with the TSS by i4C) is also methylated as part of a multi-loop structure. The same *Dpn*I sites were also tested in transfections involving a construct encoding either a non-targeting ("scrambled") TALE domain or the targeting domain fused to an inactive Dam protein ("ΔDam"). *$P < 0.05$; two-tailed unpaired Student's *t*-test ($n = 3$). The bars and error bars denote mean ± SEM.

to follow NF-κB binding (seen using either *Apo*I or *Nla*III; Appendix Fig S18A), which was further verified by differential analysis (especially at two sites in the *SAMD4A* enhancer cluster, where changes are dampened in conventional 4C; Appendix Figs S19 and S20). TNF stimulation does not change the fraction of reads mapping within the *SAMD4A* TAD (Appendix Fig S18B), many of which overlap NF-κB binding sites (Appendix Fig S18C). Then, prelooping and interaction remodeling to follow NF-κB binding (predominantly within their respective TADs) is seen for responsive TSSs, indicating these are prevalent and dynamic features of native folding.

Finally, as i4C is a "one-to-all" approach, we sought to map interactions in a global fashion. We applied a capture-based 3C method, "T2C" (Kolovos *et al*, 2014), where probes targeting every *Apo*I

fragment in our 2.8-Mbp model locus on chromosome 14 are used to retrieve and sequence a subset of 3C/i3C ligations. T2C was reproducibly performed in HUVECs to yield 1.5-kbp-resolution interaction maps (Fig 4A and Appendix Fig S21A–C). In the presence of cross-linking, all known features of genomic organization are seen; when performed natively, the outline of sub-TADs (or contact domains) is mapped at lower resolutions (Fig 4A, top), while at higher resolutions individual chromatin loops are resolved against ultra-low background (Fig 4A; bottom). In fact, using the "directionality index" approach to call TADs at 10-kbp resolution (Dixon *et al*, 2012), we find very similar organization in the presence or absence of cross-linking, with some additional subdomains emerging in iT2C (Appendix Fig S21D). Moreover, all seven CTCF–CTCF loops

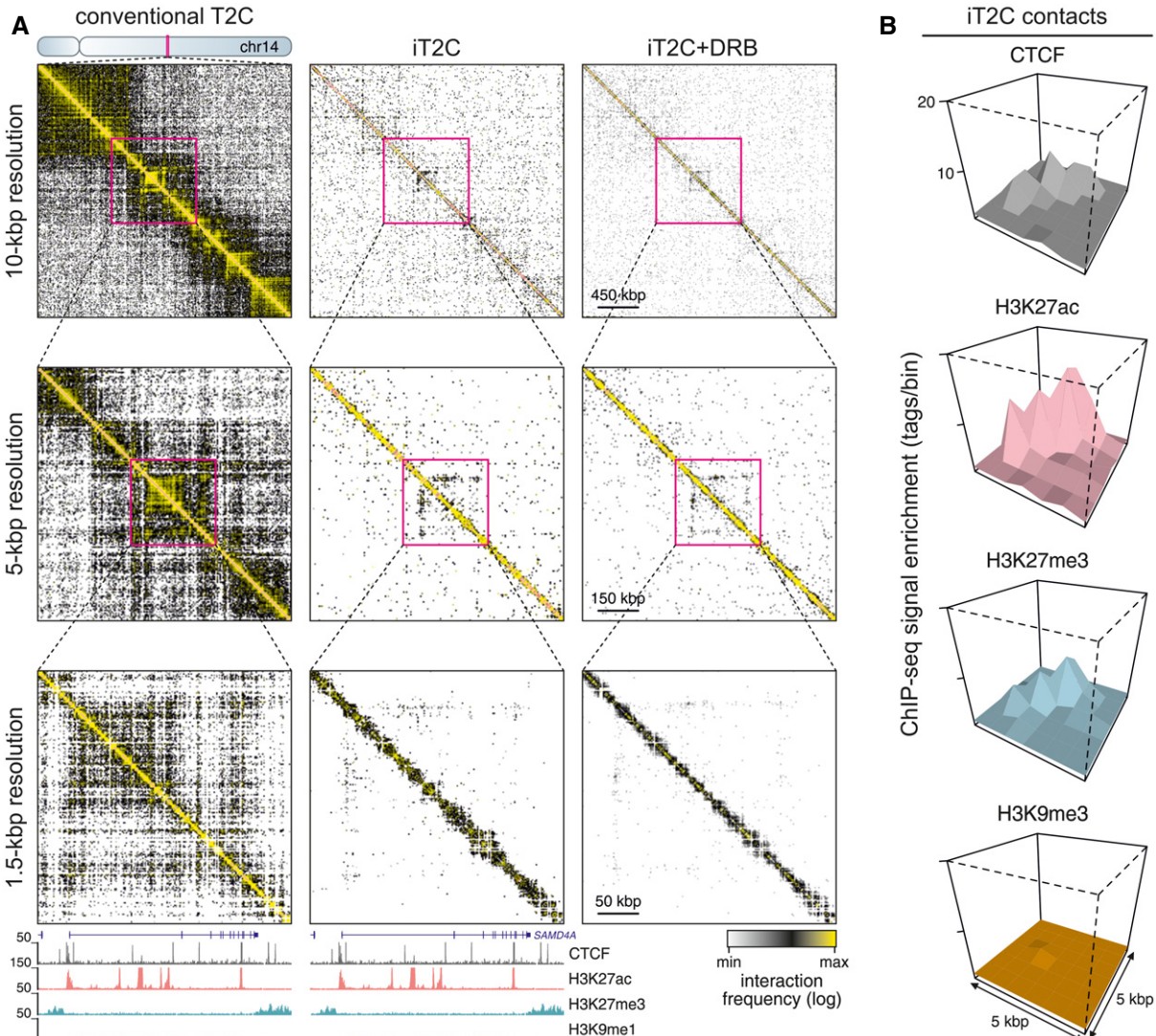

**Figure 4.    3D organization of a 2.8-Mbp human locus analyzed by iT2C/conventional T2C.**

A   Interaction maps from conventional T2C (left) and iT2C (middle, right) in the 2.8 Mbp around *SAMD4A* on chromosome 14. Magnifications show interactions at increasingly higher resolution. Bottom: HUVEC ENCODE ChIP-seq data are aligned to interactions mapped at 1.5-kbp resolution in the 250 kbp around *SAMD4A*.

B   PE-SCAN graphs (see de Wit *et al*, 2013) show the enrichment of iT2C interactions (± 5 kbp) for CTCF (gray), H3K27ac (pink), and H3K27me3 (blue), while H3K9me3 (brown) that is absent from this region serves as a control.

previously called in this 2.8-Mbp region (Rao *et al*, 2014) are picked up by iT2C (Appendix Fig S21E) and, overall, iT2C contacts are enriched for CTCF, H3K27ac, or H3K27me3 at the interacting sites (Fig 4B). Treatment of HUVECs with the transcriptional inhibitor DRB does not dramatically alter the iT2C map (e.g., prelooping in *SAMD4A* persists; Fig 4A). This supports the notion of an overarching organization that is in part independent of transcription, and was affirmed when iT2C for the same locus was applied to IMR90s (Appendix Fig S22). Importantly, iT2C is devoid of signal from "bystander/baseline" interactions, allowing us to detect contacts over ultra-low background (Appendix Fig S21F), while contact structure close to the diagonal is reminiscent of that seen by "micro-C" (Hsieh *et al*, 2015).

Finally, we provide proof-of-principle iHi-C ("all-to-all" i3C) under native conditions by using *Dpn*II, incorporating biotin-dATP in ligation junctions, and sequencing to ~$3 \times 10^8$ reads. The resulting interaction maps were compared to conventional HUVEC Hi-C (and to *in situ* Hi-C from uncross-linked agar-encapsulated lymphoblasts; Rao *et al*, 2014; Appendix Fig S23A and B). Contact strength distribution over distance is not very different in these datasets, yet iHi-C can detect CTCF–CTCF loops more robustly, and resolves individual contacts with as few as 50 million reads (Appendix Fig S23C–E). Last, in order to assess whether any contacts not captured in i3C are lost in the fraction that diffuses from nuclei upon cutting, we performed side-by-side iHi-C on the "lost" (in dilution) and "retained" (*in situ*) fractions, sequenced to > $2 \times 10^8$ reads, and compared the resulting interactions. Our analysis, exemplified by a representative region on chromosome 18, showed that very few interpretable contacts can be retrieved from the "lost" fraction, and these are often also found in the "retained" one (Appendix Fig S24).

In summary, i3C contact profiles display great similarity to conventional ones, thus alleviating most concerns about discrepancies due to fixation. Critically, our data verify the importance of topological restrictions imposed by TAD formation under native conditions, which highlights their regulatory implications *in vivo*. Moreover, the unattached chromatin lost in i3C renders captured interactions more focal, which can be advantageous when studying regions densely populated by *cis*-elements (like "super-enhancers"). Similarly, iT2C and iHi-C offer the potential to call loops at high resolution against essentially zero background without a need for excessive sequencing depth. Thus, we suggest that i3C offers a rapid and robust means for interrogating native spatial interactions; it dampens "bystander/baseline" ligations to increase signal quality, and so complements the existing toolkit for investigating 3D genome architecture in eukaryotes.

# Materials and Methods

### Cell culture

HUVECs from pooled donors (Pan Biotech.) were grown in Endothelial Basal Medium with supplements and 3% fetal bovine serum (FBS; Pan Biotech.). IMR90s (Coriell Repository) were grown in MEM (Sigma-Aldrich) with 20% FBS (Gibco) and 1% non-essential amino acids (Sigma-Aldrich). K562 cells were grown in RPMI (Sigma-Aldrich) with 10% FBS, and mouse embryonic stem cells (E14 mESCs) in Knockout-DMEM medium (Life Technologies)

containing 15% FBS and LIF (a gift by Alvaro Rada-Iglesias) on gelatin-coated plates. All cells were grown to ~90% confluency before harvesting for further processing or passaging. Where appropriate, cells were serum-starved overnight and treated with TNF (10 ng/ml; Peprotech) or with 50 μM DRB (Sigma-Aldrich) for 1 h at 37°C.

### i3C

An adapted close-to-physiological, isotonic buffer (PB; 100 mM $KCH_3COO$, 30 mM KCl, 10 mM $Na_2HPO_4$, 1 mM $MgCl_2$, 1 mM $Na_2ATP$, 1 mM DTT, 10 mM β-glycerophosphate, 10 mM NaF, 0.2 mM $Na_3VO_4$, and pH is adjusted to 7.4 using 100 mM $KH_2PO_4$; Kimura *et al*, 1999) is prepared fresh every time in nuclease-free water (Millipore MilliQ), supplemented with 25 U/ml RiboLock (Thermo Scientific) and protease inhibitors (Roche), and kept on ice throughout the procedure. Typically, $5 \times 10^6$ cells are used per experiment, harvested in 4 ml of ice-cold PB from 15-cm culture plates using a soft rubber cell scraper (Roth) on ice. Harvested cells are spun at 600 *g* (4°C, 5 min), resuspended and incubated (ice, 10 min) in 10 ml of PB supplemented with 0.4% NP-40 to release nuclei. This step is usually repeated 1–2 times (ice, 5 min), and nuclei integrity is checked on a hemocytometer. Isolated nuclei are collected via centrifugation at 600 *g* (4°C, 5 min), gently resuspended in 500 μl of ice-cold PB/0.4% NP-40, and transferred to 2-ml round-bottom, low-retention tubes. Next, chromatin is digested with 500 units of *Apo*I or *Nla*III (New England Biolabs; 33°C, 30–45 min) without shaking. Aliquots of 10 μl are put aside right before and after digestion as "uncut" and "cut" chromatin controls. Treated nuclei are then spun at 600 *g* (4°C, 5 min) to separate cut, unattached chromatin fragments that are released into the supernatant, washed in 500 μl of ice-cold PB, and respun. Following resuspension in 1 ml of ice-cold PB, spatially proximal chromatin ends are ligated together in intact nuclei (an idea based on the original "proximity ligation" assay by Cullen *et al*, 1993) and supported by recent findings that, even under cross-linked conditions, ligations predominantly occur within the "chromatin cage" of intact nuclei; Gavrilov *et al*, 2013) by adding 100 units of T4 DNA ligase (5 U/μl stock; Invitrogen) and 10 μl BSA (10 mg/ml stock; Sigma-Aldrich), and incubating at 16°C for 6–12 h without shaking. Finally, 25 μl proteinase K (10 mg/ml stock; AppliChem) are added to the samples, which are kept at 42°C overnight. Next day, samples are treated with 25 μl RNase A (10 mg/ml stock; AppliChem; 37°C, 1 h) and purified by phenol/chloroform extraction (pH 8.0) and ethanol precipitation. To reduce co-precipitating DTT, the aqueous phase volume is increased to 1 ml using nuclease-free water, 200 μl 3 M sodium acetate, and 5 ml absolute ethanol are added, and tubes are placed at −80°C for 30 min. Following centrifugation at 4,500 *g* (4°C, 1.5 h), pellets are washed in 5 ml 70% ethanol, air-dried for ~20 min at room temperature, dissolved in 70–100 μl of TE (pH 8.0) at 37°C for 20 min, and the concentration of the i3C template determined using a Qubit 2.0 Fluorometer (Life technologies).

For i4C-seq, circularization and inverse PCR were as described previously (Stadhouders *et al*, 2013); ~25 μg of i3C template are digested with 25 units of *Dpn*II (New England Biolabs; 37°C, overnight). After heat inactivation (65°C, 25 min), DNA is diluted in ligation buffer to a volume of 7 ml, religated using 20 μl T4 DNA ligase (5 U/μl stock; Invitrogen; 16°C, 6–8 h), and purified. Then,

~150 ng of the circularized i3C template is used in inverse PCRs as follows: one cycle at 95°C for 2 min, followed by 34 cycles at 94°C for 15 s/56–58°C for 1 min/72°C for 3 min, before a final extension at 72°C for 7 min using 3.75 units of the Expand long template HF DNA Polymerase (Roche). Typically, eight such PCRs are pooled, purified using the DNA Clean & Concentrator kit (Zymo Research), and amplicons checked by electrophoresis on a 1.5% (wt/vol) agarose gel. The rest of the sample are directly sequenced on a HiSeq2500 platform (Illumina) as the primers used in inverse PCRs carry the P5/P7 Illumina adapters as overhangs. All primers used in this study are listed in Appendix Table S1.

For i3C-qPCR, ~100 ng of the i3C/3C template was used. Primers were designed using Primer3Plus (www.bioinformatics.nl/cgi-bin/primer3plus/primer3plus.cgi) to have a length of 18–23 nucleotides, a Tm of 58–62°C, and to yield amplimers of 70–150 bp. qPCRs (15 μl) were performed using the SYBR Green JumpStart Ready Mix (Sigma) on a Rotor-Gene Q cycler (Qiagen; one cycle of 95°C for 5 min, followed by 40 cycles at 95°C for 15 s, 61°C for 40 s, and 72°C for 20 s). i3C/3C amplimer levels were normalized to both a "loading" primer pair (for equiloading) and to templates prepared by cutting and ligating bacterial artificial chromosomes (BACs) spanning the studied loci of interest (controls for primer efficiency). All primers are available on request; all BAC's used in this study are listed in Appendix Table S2.

### Data analysis

The analysis of high-throughput sequencing data from i4C-/4C-seq experiments was carried out using the fourSig (Williams *et al*, 2014) or FourCSeq (Klein *et al*, 2015) packages. In brief, 76-bp single-end reads from a HiSeq2000 platform (Illumina) were trimmed to remove the viewpoint primer sequence using homerTools (http://homer.salk.edu/homer/). Trimmed reads were then mapped to the reference genome (hg19) using the short read aligner BWA-MEM (Li & Durbin, 2010; exact parameters were as follows: BWA MEM -t 8 -k 15 -r 1 -B 1 –M) and processed via fourSig or FourCSeq. Data were then visualized by uploading. BedGraph files to the UCSC genome browser (https://genome.ucsc.edu/; hg19) and using its embedded smoothing option. i4C-seq replicates and mapping efficiencies are listed in Appendix Table S3, and quality control/correlation plots (as in van de Werken *et al*, 2012) shown in Appendix Fig S7D and E.

### Data availability

All i3C data generated here are available at the EBI Array Express archive under accession number E-MTAB-4719. Data generated using the conventional 3C approach are available at the Sequence Read Archive under the accession number SRP066044.

Expanded View for this article is available online.

### Acknowledgements

We thank Erez Lieberman Aiden, Alvaro Rada-Iglesias and Leo Kurian for discussions; Peter Cook and Svitlana Melnik for help during the early stages of this work; Elzo de Wit for the PE-SCAN code; Bradley Bernstein and Eric Mendenhall for the TALE plasmids; Michaela Bozukov for help with experiments; and the Cologne Center for Genomics for sequencing iHi-C libraries.

Work in AP's laboratory is supported by the Fritz Thyssen Stiftung, the DFG Excellence Initiative via a "UoC Advanced Researcher Grant", and by CMMC Junior Research Group funding; DM is supported by an ERC consolidator grant.

### Author contributions

LB and AP designed experiments and developed the method. LB performed experiments. WI sequenced i4C/iT2C libraries. PK and FGG helped with T2C experiments. TG and MN analyzed high-throughput data. CAB and DM performed simulations. LB and AP wrote the manuscript with input from all co-authors.

### Conflict of interest

The authors declare that they have no conflict of interest.

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
