## [Review Process File · Molecular Systems Biology]

Exploiting native forces to capture chromosome conformation in mammalian cell nuclei

Miss Liliya Brant, Theodore Georgomanolis, Mr. Milos Nikolic, Mr. Chris Brackley, Petros Kolovos, Wilfred van IJcken, Frank G. Grosveld, Davide Marenduzzo and Argyris Papantonis

Corresponding author: Argyris Papantonis, University of Cologne

Review timeline:	Submission date:	07 September 2016
	Editorial Decision:	27 September 2016
	Revision received:	08 November 2016
	Accepted:	14 November 2016

Editor: Maria Polychronidou

Transaction Report:

1st Editorial Decision

27 September 2016

Thank you again for submitting your work to Molecular Systems Biology. We have now heard back from the reviewers who agreed to evaluate the study. As you will see below, the reviewers appreciate that the presented approach is a valuable contribution to the field. They raise however a series of concerns, which should be carefully addressed in a revision.

I think that the recommendations of the reviewers are quite clear so there is no need to repeat the points listed below. Please feel free to contact me in case you would like to discuss any of these points in further detail.

REFeree REPORTS

Reviewer #1:

By comparing the 3C, 4C Hi-C results from native and crosslinked cells, they showed that most of the chromatin interactions are conserved in the absence of x-linking. In other words, most of the previously discovered chromatin interactions from formaldehyde crosslinking cells are reliable. This is an important contribution to the field as skepticism about 3C methods persist.

Another paper had previously reported Hi-C with un-crosslinked cells but seemed to have higher background. Here, the authors have done a lot of experiments with the un-crosslinked condition native 3C: i3C, i4C, iHi-C and iT2C. The results and methods will be useful to the field. It would be good to get more information about what interactions are different between x-linked and non x-linked methods. Are there any general features (e.g. pol2 occupancy, histone marks etc that would be predictive of differences?

Also, the TALE-iD technology is very interesting. This method could be developed further and published separately.

A few minor comments:

1. In Fig.EV1B, all the enzymes that didn't work are 6-bp cutters. Is this a coincidence or because of other reasons?
2. Fig.3C, why does the scrambled TALE also have enriched signal around the anchor?
3. Some of the names are not consistent. In Fig.EV8, the second track from top, does "i3C-seq" mean "i4C-seq" as used in other places? Also, "TALE-iD" in Fig.3A and main text, but "TALEN-iD" in the legend.
4. Fig.EV17D, switching site of the directionality index (DI) is not equal to TAD boundary. The DI also switches in the middle of TADs. To get the boundaries, the authors need to run HMM after calling DI.

Reviewer #2:

Brant et al. describe an improved technique to study DNA:DNA interactions at a global scale within individual nuclei without the need for chemical crosslinkers. They show that their technique provides data of similar quality to 3C or 4C and corroborates the results observed using 3C and 4C-based approaches. They show that their approach can be applied to all:all comparisons such as those in HiC, and provides results comparable to high-resolution modifications of the technique such as micro-C. They have two main conclusions: first, that the chemical biases inherent in the crosslinker do not significantly change the patterns of association discovered by these methods, and second, that crosslinkers may not be necessary for most studies of genome organization.

The authors provide a thorough analysis of the applicability and robustness of their technique. The description of a crosslink-free method is of significance and the observation of similar results to approaches that use crosslinking is reassuring and addresses a persistent question in the field. As such this study is a very valuable contribution. We have only few major points of criticism but we would like clarification of some.

Major points:

- 1) The authors observe that more than 40% of chromatin is lost after digestion and before sequencing. They claim that this represents nonspecific loss and use as evidence for this claim that there is relatively similar enrichment for various annotations in the lost and retained fractions. We would like some more information to clarify whether nuclear structure or DNA sequence contribute to whether a fragment is lost or retained. For example:
 - a. Does the size of an ApoI fragment contribute to its likelihood to be retained or lost?
 - b. Are known structural elements such as LADs or Nucleolar Organizing Regions enriched in one pool of fragments?
 - c. If a locus has mapped reads from the retained pool, is it more or less likely to also be present in the lost pool (are these non-overlapping sets)?
- 2) The authors observe a striking sparsity in the iT2C data, as compared to the conventional T2C data they present. We are curious what the sources, and consequences, of this are. In particular:
 - a. Do the two libraries have similar coverage, and similar QC scores (total reads, percent mapped without errors, percent duplicated etc)? If this is not the case, what does the conventional T2C data look like when you resample it down to levels comparable to the iT2C data (or vice versa)?
 - b. In the i3C data, the authors noted that a full 83% of sequenced reads came from loci with more than 100 reads per million. Is this also true of the iT2C data, and if so is it causing the observed sparseness? (If this is the case, could the graphs in figure 4A be renormalized to better reflect this?)
 - c. What is the signal:noise ratio in iT2C data as opposed to T2C data? Can this be quantified using the previously identified CTCF-CTCF loops?
- 3) In general, can the authors include in their supplemental material and in their materials and methods section a more thorough discussion of the number of reads sequenced, number of reads mapped, and quality control metrics performed on these reads?

Minor points:

- 1) The authors use the terminology "more focal" to describe their data without specifying precisely what they mean by this. Please provide a specific metric for this observation.
- 2) The authors observe that 40% of cis-contacted fragments are shared between traditional 4C and i4C. This is surprisingly low. Are most of the 60% non-overlapping cis-contacts found only in 4C data or only in i4C data?
- 3) Similarly, in light of this, how correlated are i4C and conventional 4C?
- 4) Could the authors provide some quantification of the extent to which i4C shows "significantly lower numbers of "uncut" and "self-ligation" reads... as well as more reads mapping within its TAD"? What is the overall percent change and what is the level of significance?
- 5) Could the authors specify how the i3C-based techniques specifically differ from in situ Hi-C without a crosslinker?
- 6) Using conventional 4C, can small changes in structure upon NF- κ B binding be observed, or are these observations only present in i4C?

Reviewer #3:

The manuscript describes a modification to standard #C-related technology to carry out this procedure without formaldehyde fixation. The authors present solid evidence showing that the new procedure gives equivalent results than fixation-based methods, but with some notable advantages. Overall, the manuscript represents an important contribution to this rapidly moving field. There are a few concerns the authors should address:

1. Step 3 in the iC technique eliminates some DNA by centrifugation of nuclei. Figure 1C indicates that the material eliminated contains enhancers, promoters, insulators, etc. The authors should show more convincingly that this material does not represent important contacts.
2. In Figure 2A, I believe the authors show domains obtained from Rao et al. and refer to these domains as TADs. However, the domains identified using the Arrowhead algorithm are not TADs, rather, they are smaller domains referred to by Rao et al as "loops" or contact domains, which are smaller than TADs.
3. In supplementary Figure EV11 A, it appears that treatment with RNase I results in a general decrease in interaction frequency that the authors interpret as evidence that RNA plays an important role in maintaining interactions. An alternative explanation is that RNase treatment increases the amount of material release from nuclei in step 3 of the procedure, and that the effect is not specific.
4. In Figure EV19 B, it appears that all the compartmental interactions normally found in HiC disappear in the iHi-C procedure. The authors should comment on this. Authors should also comment on the multiple lines, some of them perpendicular to the diagonal, that appear in these figures. Do these lines correspond to rearrangements in the cells used for the experiments?

1st Revision - authors' response

08 November 2016

Text continued on next page.

Point-by-point response to reviewers' comments

Reviewer #1:

By comparing the 3C, 4C Hi-C results from native and crosslinked cells, they showed that most of the chromatin interactions are conserved in the absence of x-linking. In other words, most of the previously discovered chromatin interactions from formaldehyde crosslinking cells are reliable. This is an important contribution to the field as skepticism about 3C methods persist.

Another paper had previously reported Hi-C with un-crosslinked cells but seemed to have higher background. Here, the authors have done a lot of experiments with the un-crosslinked condition native 3C: i3C, i4C, iHi-C and iT2C. The results and methods will be useful to the field. It would be good to get more information about what interactions are different between x-linked and non x-linked methods. Are there any general features (e.g. pol2 occupancy, histone marks etc that would be predictive of differences? Also, the TALE-iD technology is very interesting. This method could be developed further and published separately.

We thank the reviewer for finding that our “*results and methods will be useful to the field*”. As regards features that would be predictive of differences, there are no obvious ones, but it is safe to say that the absence of recognizable features at particular contacts is only applicable to the conventional method. These mostly represent “bystander” or “baseline” interactions and are simply absent from i3C data. Of course, we cannot rule out that some hitherto unknown mark/feature might support such differences. Finally, as regards the TALE-iD approach, we thank the reviewer for highlighting it. We are indeed trying to develop a more global variant of it, but this is very much still work in progress.

A few minor comments:

1. In Fig.EV1B, all the enzymes that didn't work are 6-bp cutters. Is this a coincidence or because of other reasons?

This is a coincidence. There are 4-bp cutters (e.g., *MboI*) that also do not work in our PB buffer and it is due to incompatibility with our “physiological” buffer’s composition; for example, the pre-2013 batches of *HindIII* by New England Biolabs worked fine in PB, but the new batches that followed require different salts/pH/etc and simply do not cut efficiently.

2. Fig.3C, why does the scrambled TALE also have enriched signal around the anchor?

We thank the reviewer for prompting us to look into this. It seems that the primer pair targeting region “p1” was somewhat “overefficient”; we have now replaced this particular pair, and added some more targeting other *DpnI* sites in the *ZFPM2* locus, showing that no such trend exists.

3. Some of the names are not consistent. In Fig.EV8, the second track from top, does "i3C-seq" mean "i4C-seq" as used in other places? Also, "TALE-iD" in Fig.3A and main text, but "TALEN-iD" in the legend.

We thank the reviewer for pointing out these oversights so we could correct them throughout the text; the correct terms are “i4C-seq” (not “i3C-seq”) and “TALE-iD” (not “TALEN-iD”).

4. Fig.EV17D, switching site of the directionality index (DI) is not equal to TAD boundary. The DI also switches in the middle of TADs. To get the boundaries, the authors need to run HMM after calling DI.

This is correct, and we did also apply such a Hidden Markov Model. However, at the resolution of T2C (10-kbp or less) the sub-domains called essentially coincide with the switching sites of the DI.

Reviewer #2:

Brant et al. describe an improved technique to study DNA:DNA interactions at a global scale within individual nuclei without the need for chemical crosslinkers. They show that their technique provides data of similar quality to 3C or 4C and corroborates the results observed using 3C-based approaches. They show that their approach can be applied to all:all comparisons such as those in HiC, and provides results comparable to high-resolution modifications of the technique such as micro-C. They have two main conclusions: first, that the chemical biases inherent in the crosslinker do not significantly change the patterns of association discovered by these methods, and second, that crosslinkers may not be necessary for most studies of genome organization. The authors provide a thorough analysis of the applicability and robustness of their technique. The description of a crosslink-free method is of significance and the observation of similar results to approaches that use crosslinking is reassuring and addresses a persistent question in the field. As such this study is a very valuable contribution. We have only few major points of criticism but we would like clarification of some.

We would like to thank the reviewer for finding that we “provide a thorough analysis of the applicability and robustness” of i3C, and that “this study is a very valuable contribution”.

Major points:

1) The authors observe that more than 40% of chromatin is lost after digestion and before sequencing. They claim that this represents nonspecific loss and use as evidence for this claim that there is relatively similar enrichment for various annotations in the lost and retained fractions. We would like some more information to clarify whether nuclear structure or DNA sequence contribute to whether a fragment is lost or retained. For example:

a. Does the size of an Apol fragment contribute to its likelihood to be retained or lost?

No, this does not seem to be the case. We used our sequencing data from retained and lost fragments after cutting with *NlaIII* (we have not sequenced *Apol*-cut chromatin), and identified the ~300,000 that are most enriched (more than 5-fold) in each fraction compared to the other. The size distributions were essentially identical between the two groups (with a median of ~275 bp), despite the broad range of fragment sizes (50-4000 bp). This information is now added to **Figure S1**.

b. Are known structural elements such as LADs or Nucleolar Organizing Regions enriched in one pool of fragments?

As above, this does not seem to be the case. We used the genomic coordinates of LADs and NADs from fibroblasts (unfortunately there exists no such dataset for HUVECs), and overlapped them with the most enriched ~300,000 fragments in the “lost” or “retained” fractions (as above). Looking at either the number of reads carried by the fragments in each fraction or their sizes, no significant difference is observed. This is also now included in **Figure S1**, and points to the lost and retained pools being overlapping sets (see also below).

c. If a locus has mapped reads from the retained pool, is it more or less likely to also be present in the lost pool (are these non-overlapping sets)?

The reviewer raises a very good point. Indeed, the “lost” and “retained” fragment pools are sets that overlap by almost 70%, and very few fragments are consistently depleted from one fraction while always being present in the other. We exemplify this by comparing the two pools in two chromosomes that vary greatly in size and gene content – chromosomes 2 and 20. Our analysis (now included in **Figure S1**) shows that the distribution of fragments unique to each pool in respect to HUVEC ChromHMM segments is very similar. There is only a small de-enrichment of the

“retained” pool for repetitive elements, and a slight enrichment for “active promoters” and “strong enhancers”. Overall, this points to a heterogeneity that should probably be expected of a cell population, and is in line with the conclusions drawn from single-cell Hi-C (Nagano *et al*, 2013).

2) The authors observe a striking sparsity in the iT2C data, as compared to the conventional T2C data they present. We are curious what the sources, and consequences, of this are. In particular:

a. Do the two libraries have similar coverage, and similar QC scores (total reads, percent mapped without errors, percent duplicated etc)? If this is not the case, what does the conventional T2C data look like when you resample it down to levels comparable to the iT2C data (or vice versa)?

The conventional and iT2C datasets display very similar coverage (~45 million read pairs each), mapping efficiencies (~65%), and content of duplicates (<10%; see **Table S4** for details). Thus, no resampling is needed – the sparsity is not at all due to technical discrepancies, but rather an inherent feature of the i3C method.

b. In the i3C data, the authors noted that a full 83% of sequenced reads came from loci with more than 100 reads per million. Is this also true of the iT2C data, and if so is it causing the observed sparseness? (If this is the case, could the graphs in figure 4A be renormalized to better reflect this?)

We have added the relevant graph for the iT2C data (in comparison to conventional T2C) in **Figure S21**. Here, the conventional approach has ~90% interactions carrying <10 reads per million, whereas iT2C only ~60%. This is in agreement with a loss of “bystander/baseline” interactions from the iT2C data and explains the sparse motifs in the interaction matrices. Nonetheless, the data in **Figure 4A** were already presented normalized and plotted at the same scale. Hence, the matrices shown are not skewed as a result of these differences (e.g., **Figures S21** and **S22**).

c. What is the signal:noise ratio in iT2C data as opposed to T2C data? Can this be quantified using the previously identified CTCF-CTCF loops?

We have approached this calculation in different ways, but, because of the virtually empty matrix surrounding CTCF-CTCF contact bins (giving a denominator value close to 0), the ratio calculated is probably overstated. Thus, unfortunately, we cannot offer a numerical metric, but we believe that the matrices compared, for example, in **Figure S21F** exemplify this signal-to-noise improvement.

3) In general, can the authors include in their supplemental material and in their materials and methods section a more thorough discussion of the number of reads sequenced, number of reads mapped, and quality control metrics performed on these reads?

We apologize for the oversight. Although these metrics were included for all i4C experiments, the iT2C data were left out – we now include these as **Table S4**, and have also added extra details on quality control metrics for NGS reads and their mapping in the **Appendix Methods** section.

Minor points:

1) The authors use the terminology “more focal” to describe their data without specifying precisely what they mean by this. Please provide a specific metric for this observation.

The reviewer is right in pointing out the lack of a specific metric. We now provide this in **Figure S6** by comparing the breadth of interactions called by the *foursig* algorithm in all i4C or conventional 4C data generated using the *SAMD4A* TSS as a viewpoint. Both the distribution and the median size of contacted fragments are significantly smaller in i4C.

2) The authors observe that 40% of cis-contacted fragments are shared between traditional 4C and i4C. This is surprisingly low. Are most of the 60% non-overlapping cis-contacts found only in 4C data

or only in i4C data?

The 40% overlap in *cis*-contacted fragments between conventional 4C and i4C refers to the analysis of fragments carrying >100 rpm (in an effort to focus on “stronger” interactions). If we look at the overlap of all *cis*-contacted fragments the number will increase to >65% — still, we must note that the vast majority of differences is found at fragments outside the viewpoint’s TAD.

3) Similarly, in light of this, how correlated are i4C and conventional 4C?

Pairwise correlations of i4C and conventional 4C datasets (and also between i4C replicates) are shown in **Figure S5B** and **S7D**; for these all Spearman’s correlation coefficients are typically >0.7.

4) Could the authors provide some quantification of the extent to which i4C shows "significantly lower numbers of "uncut" and "self-ligation" reads... as well as more reads mapping within its TAD"? What is the overall percent change and what is the level of significance?

For uncut and self-ligation reads, the levels drop from 4-5% in conventional 4C to <2% in i4C (often to <1%). For reads mapping within TADs, the levels rise from ~15% in conventional 4C to >20% in i4C, while we observe ~10% fewer *trans*-contacted fragments. All these changes are statistically significant (*P*-value <0.01; Student’s unpaired t-test), and the data is included in **Figure S6**.

5) Could the authors specify how the i3C-based techniques specifically differ from in situ Hi-C without a crosslinker?

We have tried to address this in two ways. First, we tried to reproduce the uncrosslinked *in situ* Hi-C protocol to generate 4C data from (the description in the original Rao *et al* paper only states that the procedure is identical to the crosslinked *in situ* Hi-C, but “with more gentle handling”; this means that both heating is used and SDS is added to uncrosslinked nuclei). In our hands, this did not work efficiently, even with short exposures to heating and lower SDS titers. Nonetheless, we did produce some 4C-seq data for the *SAMD4A* viewpoint (**Figure S13**), which when compared to the i4C approach seem markedly depleted of major interactions. We attribute this to the harsher handling of the *in situ* protocol, and also to the lack of a physiological buffer. Second, we looked at the published Hi-C data generated without crosslinking (Rao *et al*, 2014). In their paper, Rao *et al* generated 5 such libraries from uncrosslinked lymphoblasts; 4 of these were on cells embedded in agar plugs, and 1 in agar-free cells. Although this last library is the one that mostly resembles our approach, the data therein is sparse. Thus, wanting to do this nice work justice, we only compare our iHi-C to data from Hi-C on agar-embedded cells (where cut chromatin cannot escape nuclei). This led to three main observations: (1) the large-scale, low-resolution, structure of A/B compartments is sufficiently retained in agar-embedded cells, but is very faint in iHi-C; (2) at high-resolution the agar-embedded maps are more noisy and only few strong structural features stand out; (3) CTCF-CTCF loops are more difficult to detect in the agar-embedded Hi-C data, in contrast to the high local enrichments seen in iHi-C (**Figure S23**). These observations are now added to the main text (pg. 8-9), and we believe they highlight how the two approaches differ.

6) Using conventional 4C, can small changes in structure upon NF-κB binding be observed, or are these observations only present in i4C?

The vast majority of spatial interactions seen by conventional 4C in three loci on chr14 (*BMP4*, *CDKN3*, and *SAMD4A*; see **Figure S17**) remain unchanged upon TNF stimulation. This appears to be less so when using i4C for the same viewpoints, where some distinct changes (that also match the regulation of these three loci) were detected. We exemplify this by directly comparing raw profiles (**Figures S16** and **S17**) and by performing differential analyses of interactions (**Figure S19** and **S20**).

Reviewer #3:

The manuscript describes a modification to standard Hi-C-related technology to carry out this procedure without formaldehyde fixation. The authors present solid evidence showing that the new procedure gives equivalent results than fixation-based methods, but with some notable advantages. Overall, the manuscript represents an important contribution to this rapidly moving field.

We would like to thank this reviewer for acknowledging that our work “represents an important contribution to this rapidly moving field”.

There are a few concerns the authors should address:

1. Step 3 in the i3C technique eliminates some DNA by centrifugation of nuclei. Figure 1C indicates that the material eliminated contains enhancers, promoters, insulators, etc. The authors should show more convincingly that this material does not represent important contacts.

As this is a concern raised by both reviewers, we tried to address it in a number of ways, on top of the analysis that was already included in the manuscript. We have now added new analyses (**Figure S1**), i3C-qPCR validations (**Figure S3**), plus an iHi-C experiment comparing contacts in the “lost” and “retained” fractions” (**Figure S24**); they all support the notion that few (if any) “meaningful” contacts are specific to the material lost upon chromatin cutting of uncrosslinked nuclei.

2. In Figure 2A, I believe the authors show domains obtained from Rao et al. and refer to these domains as TADs. However, the domains identified using the Arrowhead algorithm are not TADs, rather, they are smaller domains referred to by Rao et al as “loops” or contact domains, which are smaller than TADs.

The domains showed in **Figure 2A** (and elsewhere, where domains are outlined on top of Hi-C data from Rao *et al*) were identified using the “directionality index/HMM” approach at 40-kbp resolution data (Dixon *et al*, 2012). This is now clarified in the corresponding figure legends.

3. In supplementary Figure EV11A, it appears that treatment with RNase I results in a general decrease in interaction frequency that the authors interpret as evidence that RNA plays an important role in maintaining interactions. An alternative explanation is that RNase treatment increases the amount of material release from nuclei in step 3 of the procedure, and that the effect is not specific.

The reviewer is right in pointing this possibility out. We have now performed an experiment to test this: we apply i3C to HUVECs by cutting chromatin with *Apo1* and post-treating (or not) with RNase A. Then, using a Qubit device, we find a ~1.5-fold increase in the amount of released chromatin; this is now added to the manuscript (pg. 6 and **Figure S13E**).

4. In Figure EV19B, it appears that all the compartmental interactions normally found in HiC disappear in the iHi-C procedure. The authors should comment on this. Authors should also comment on the multiple lines, some of them perpendicular to the diagonal, that appear in these figures. Do these lines correspond to rearrangements in the cells used for the experiments?

Indeed the large-scale compartments typically seen in Hi-C are not observed in iHi-C (and this is not at all connected to sequencing depth or the resolution at which the data are portrayed). We now point this out in the main text. As regards the lines perpendicular to the diagonal, these are also a feature of the conventional Hi-C, not an aberration of iHi-C. They are seen in multiple loci across the genome and typically represent interactions between large H3K27me3-marked regions. Thus, they do not represent genomic rearrangements in HUVECs (in fact, we also see them in IMR90s, in the agar-embedded lymphoblasts assayed by Rao *et al*, and in conventional Hi-C from three single HUVEC donors; *own unpublished data*).

Thank you again for sending us your revised manuscript. We are now satisfied with the modifications made and I am pleased to inform you that your paper has been accepted for publication.

Corresponding Author Name: Argyris Papantonis

Manuscript Number: MSB-16-7311